# An Analysis into Physical and Virtual Power Draw Characteristics of Embedded Wireless Sensor Network Devices under DoS and RPL-Based Attacks

**DOI:** 10.3390/s23052605

**Published:** 2023-02-27

**Authors:** Patryk Przybocki, Vassilios G. Vassilakis

**Affiliations:** Department of Computer Science, University of York, York YO10 5GH, UK

**Keywords:** wireless sensor network, RPL, power draw attack, DoS attack

## Abstract

Currently, within the world, cybercrime is becoming increasingly rampant—often targeting civil infrastructure like power stations and other critical systems. A trend that is being noticed with these attacks is their increased use of embedded devices in denial-of-service (DoS) attacks. This creates a substantial risk to systems and infrastructures worldwide. Threats to embedded devices can be significant, and network stability and reliability can suffer, mainly through the risk of battery draining or complete system hang. This paper investigates such consequences through simulations of excessive loads, by staging attacks on embedded devices. Experimentation within Contiki OS focused on loads placed on physical and virtualised wireless sensor network (WSN) embedded devices by launching DoS attacks and by exploiting the Routing Protocol for Low Power and Lossy Networks (RPL). Results from these experiments were based on the metric of power draw, mainly the percentage increase over baseline and the pattern of it. The physical study relied on the output of the inline power analyser and the virtual study relied on the output of a Cooja plugin called PowerTracker. This involved experiments on both physical and virtual devices, and analysis of the power draws characteristics of WSN devices with a focus on embedded Linux platforms and Contiki OS. Experimental results provide evidence that peak power draining occurs with a malicious-node-to-sensor device ratio of 13-to-1. Results show a decline in power usage with a more expansive 16-sensor network after modelling and simulating a growing sensor network within the Cooja simulator.

## 1. Introduction

The field of wireless sensor network (WSN) devices has been growing exponentially over the last 5 years with a predicted 30.9 billion devices connected by 2025 [1]. Many important questions are being answered by the industry and research community. These include the development of protocols and technologies such as RPL [2,3,4] and 6LoWPAN [5,6] to intelligent intrusion detection solutions [7,8,9]. The trend of increasingly interconnected networks is likely to progress, with more serious attacks involving Internet of Things (IoT) devices predicted [10]. An example of a serious attack like this is the Mirai botnet [11]. In late 2016, it helped cement IoT devices within the security spectrum [12]. The Mirai botnet managed to clock one of the highest bandwidths of any distributed denial-of-service (DDoS) attack since the beginning of the internet at 1.35 Tbps [13]. It has not been demonstrated within the literature that attacks like this focus on battery draining; however, there is always a possibility and attacks like this are being actively researched [14,15,16].

Society is currently experiencing a so-called “industrial revolution” of the IoT, a topic that has been heavily explored by Lasi et al. [17]. Devices that are part of this revolution deal within all sections of manufacturing, with the growing chance of appearing within nuclear facilities or other critical infrastructure. Some facilities currently utilising networks that implement these critical infrastructure technologies are referred to as supervisory control and data acquisition infrastructures (SCADA) [18,19]. Popular SCADA solutions’ manufacturers are known to have outdated security measures implemented within their wide area sensor networks as explored by Samtani et al. [20]. With the added current events in Ukraine, any possible nuclear accident is synonymous with danger. Multiple nuclear power stations are currently housed within the borders of the country, and attacks have already been launched against similar targets [21]. Similar SCADA infrastructures with known security compromises are likely to be situated worldwide.

Due to the uncertain geopolitical nature of the world, increasing counts of cyber warfare are being recorded within critical industries like power, gas, and electricity [22,23]. Within this aforementioned conflict, drone systems and unmanned aerial vehicles (UAV) infrastructures have been heavily relied on [24]. As active systems, these embedded devices rely on battery power to communicate and operate. Being standalone systems, the devices also need to communicate wirelessly and introduce many attack vectors to UAV systems and networks [25]. The work of exploring the options of such devices has been carried out by Shi et al. [26], who studied low-power protocols that could be appropriate for systems of this nature.

With the cyber warfare issues disclosed, devices on the IoT spectrum are often taken advantage of within networks from multiple angles. Devices are often attacked, hijacked, and formed into botnets of millions of devices [27,28]. For example, the Mirai botnet [11] automatically compromised devices and acted as a worm, attacking more devices as it spread. Another example is the Chalubo botnet [29] which acted on a default password exploit and similarly to Mirai created a botnet. Attacks utilising these armies of machines are well known and have been used to cause chaos, sometimes being used to take down targets such as GitHub [13]. The GitHub attack has been recorded as one of the largest DoS attacks in history at over 1.35 Tbps. Similarly, an attack that targeted a hosting company which hosted the likes of Reddit, Etsy, and Spotify was also taken down in 2016 [12]. A more recent example of a DoS attack and one which shows how quickly attacks like this are increasing in complexity and power is the attack that hit Microsoft’s Azure web platform in November of 2021 [30]. The attack more than doubled the throughput of the previously mentioned GitHub attack at an incredible 3.47 Tbps. However, this time the attack was successfully mitigated by the platform’s DDoS protection.

Whilst the hypothetical negative effects of battery draw attacks are well documented [14,15], there are very few studies that investigate in practice the effects of such DoS attacks against WSN embedded devices. This study focuses on analysing the power draw characteristics of wireless sensor network devices running embedded Linux and Contiki operating systems. Two experiment routes are undertaken, one physical and one virtual, both on embedded platforms. The virtual study focuses on a range of RPL-based attacks against a growing network of sensor nodes. The physical study focuses on DoS attacks against a variety of operating systems. The aim is to analyse the power draw characteristics of such unforeseen circumstances within networks and to draw conclusion on the risks they pose. Both sides of experimentation explore attacks that aim to disrupt or interrupt the normal operation of WSNs. In particular, the contributions of this work are as follows:Investigate the creation, compilation, and setup of multiple operating systems suited for a physical embedded system along with the creation of a standard measurement which will enable a comparison vector for the study;Demonstrate if operating systems differ in reaction to a selection of DoS attacks by analysing the results created from the study;Investigate and virtualise common attack concepts within RPL networks against WSN devices. This includes implementing three RPL-based attacks: rank attack, SYN flood attack, and version number attack;Investigate if the enlargement of a network increases the power draw of an RPL-based network attack by increasing the network size from 10 nodes to 13 and 16.

The remainder of the paper is organised as follows: Section 2 provides analysis of background issues of WSN devices alongside research propositions and findings; Section 3 discusses the background behind the choices of devices and operating systems for the study; Section 4 provides the experimental setup and explains attacks that were investigated and the exact steps needed to replicate the study’s results. It also provides the reasoning as to why some choices were made; Section 5 and Section 6 provide our results and a discussion on our main findings. Section 7 provides final conclusions and takeaways from the paper and summarises the main findings alongside recommended future directions.

## 2. Related Work

A limited selection of works exists tackling the problem of power draw characteristics in DoS attacks against embedded WSN devices. Most studies focus on aspects like battery consumption of devices within normal operational environments [31,32]. Within this, there is an even smaller number of studies that focus on physical power analysis of attacked devices [33]. The work presented by Jung et al. [33] focuses on the detection of compromised devices and the study presents a successful algorithm which helps discover compromised devices based on projected and actual power consumption. The methodology revolves around the utilisation of an off-the-shelf device: a high end power analyser called Monsoon which can analyse power draw at an incredibly high rate of 5 GHz. The team focused on the use of this technology to analyse the power draw of each device to create large baseline data sets that provide a reference point to baseline operation. The baseline operation is calculated by analysing a predefined amount of time during which normal communication/computation happens. This method of analysis is closely replicated within our work, with an average power draw calculated over a set period being used in the study. This likely provides accurate results as it works with an average power draw, not a peak.

A similar concept of calculating a baseline power draw is also used by Grey et al. [32]. They estimated the average power use of IoT devices like the Phillips Hue, to accurately predict performance of many IoT technologies. Power consumption was based on power usage per data packet sent and was calculated from the power usage figure claimed by the manufacturer, divided by the number of packets sent during the active time. This was used to create an average power consumption per device that could be modelled and scaled easily. However, there is some doubt in the accuracy within this model compared to the model employed by Jung et al. [33]. For example, devices are not likely to consume the same amount of energy per packet, as every packet can vary in size or content. A paper tackling similar power draw and IoT performance metrics like that of Grey et al. [32] is Lauridsen et al. [31]. Here, the methodology is similar to that of Jung et al. [33], sharing the set time frame allocation framework to calculate power draw. There will likely be concrete reasons to use a particular methodology, especially because these three papers are similar. Our paper follows within the footsteps of previous work [31,33] as the area of focus appears to be similar.

A large issue preceding the direct comparison from the power draw of an idling device to a device that is being attacked is the unrealistic usage scenario that that would create. Devices within industries are likely to always have some sort of processing duties (in between standby/low power mode cycling). A few realistic use case scenarios could be based on initially connecting to the gateway and initialising the routing for the network, and then a series of requests can be cycled to provide an update to the user. Multiple realistic usage scenarios have been outlined within research ranging from air quality monitoring systems [34] to critical health devices [35]. With multiple ranging baselines throughout the literature, average use is likely all dependent on the frequency of packet transfer of the devices. However, some sources claim 10 years at a data transmission rate of 128 bytes per 10 s [36] and others claim 306 h [37] on one charge with frequent data transfer (as it was a vital sign monitoring system). Likely, an in between of this would be the best choice for an appropriate baseline power draw for the study, increasing the comparability of the study’s results to real-world examples where a default operation time has already been established. Basing the impact of any possible attacks on the decrease of operational time would likely result in an unbiased proposal for ranking the effectiveness of each propositioned attack.

## 3. Methodology: Operating System and Hardware Selection

This section presents our adopted methodology. In particular, we describe and compare the operating systems and hardware options available for this study. We also discuss the realistic use cases for appropriate baseline power consumption.

### 3.1. Operating Systems

In our considered scenarios, Contiki-NG would be the top choice; however, its relatively small compatibility with systems that are not micro-controller based hinders its adoption. Contiki-NG was utilised within the simulated section of experimentation; however, it is not the best choice for a physical study. This fact allows for the consideration of several other operating systems that will be compatible with other instruction sets. It is also especially important to diversify the hardware that can be used, as discussed in Section 3.2.

In our experiments, we decided to take forward three operating systems that differ from each other and add another vector of comparison to the study, whilst bringing more data to the research community. The three chosen are: Tiny Core Linux, which is essentially a bare-bones Linux installation that is not IoT specific and features a graphical user interface (GUI); Yocto OS, which is an IoT focused distribution with a minimal footprint; and Raspberry Pi OS, which also depends on a GUI-based platform. The latter, however, is based on Debian which is not known for running on low power hardware. This will likely provide a harsh contrast to how a fully-fledged operating system reacts to a similar attack. We studied the available operating systems and created Table 1 to showcase key areas of operating systems and to provide a basis for cross-comparison of the options available. This comparison is based on the architecture support, community support, low-power protocol support, kernel used, and complexity of installation.

### 3.2. Hardware

In this subsection, we describe the considered hardware, covering Intel, Raspberry Pi, and Zolertia options.

Intel. Edison (discontinued): A micro sized computing module that was proposed as competitor to other system-on-chip (SoC) options. The device is a secure digital (SD) card sized SoC aimed at creating direct competition with Arduino and Raspberry Pi offerings. Edison utilised an Intel Atom Silvermont-based chip created to compete with ARM-based micro-architectures. It is based on a 32-bit system [38] and was discontinued in 2017 [39].

Raspberry Pi Pico: The Pico is based on a micro controller architecture, making the device incredibly accessible. However, it does have a few limitations due to this: a lack of networking capabilities is one of them [40]. Due to the lack of this feature, many modification kits exist to enable networking on the Pico. An example of such can be sourced from Pimorini [41]. Another option is to utilise Mistry’s [42] ’Pico networking framework’, which bases its operation on an off-the-shelf component that can be connected to the device through its various I/O ports. The system creates a virtual Ethernet MAC stack within software. It incorporates an implementation of lwIP protocol suite to handle Ethernet connectivity. However, this option under-clocks the Pico to 50 MHz to match the clock of the Ethernet port and does not support many IPv4 protocols. Zero and Zero 2 W: Are essentially fully-fledged computers with Linux-based operating systems [40]. These systems are incredibly popular with DIY projects offering great aftermarket support. The Zero is ARMv6 based and uses a 32-bit architecture, whereas the Zero 2 is ARMv8-A based and uses a 64-bit architecture. The Zero 2, however, is unfortunately largely unavailable due to the post-pandemic chip shortage. Both devices feature a 40-pin general purpose input/output (GPIO) interface fully accessible within the software layer [40], allowing for multiple vectors of use within the industrial or hobby sector.

Zolertia Z1: Is a very popular option within research fields and is featured as a device within a Cooja simulator [43]. It is an ARM Cortex-based device. However, it is now discontinued and unavailable to buy anywhere. Zolertia Firefly: Is a possible successor to the Z1; it is also ARM Cortex-based and can be purchased from Zolertia themselves. The device retails in the region of 50€ and are featured within IoT-Lab [44] and Cooja.

When considering price and performance, the choices fall into the Raspberry Pi family. However, to replicate the hardware as closely as possible, a micro controller-esque device would be a closer match. A Zolertia Firefly is a good option for the study, although they are very difficult to source and is a slightly more expansive option. In addition, Firefly would require additional communication hardware to interface with it (a JTAG-ulator for example). A Pico with a hard-wired Ethernet port (networking) is a much cheaper device; however, software-wise, it unfortunately falls short, largely due to it not supporting any stand-alone operating systems. Due to this being a custom device, there is a high likelihood of device malfunction which could set the study back. The Raspberry Pi Zero is the best option second to the modified Raspberry Pi Pico. The device is still based on a 32-bit architecture with low power draw and aligns itself with similar embedded devices. Another large plus of the proposition is its low price and large availability. Table 2 summarises the hardware options available. This comparison is based on CPU architectures, community support, wireless connectivity support, price, availability on the market, any extras needed such as additional SD cards or software, and finally the support for our selected operating systems. From the hardware devices initially identified, there are many that fit the criteria of the study, often even including wireless communication possibilities and large community support to provide help. However, a large consideration is the price of these devices and the compatibility with the software options chosen, which are the two largest criteria within this comparison. The devices that appear to be both well priced and compatible with the operating systems chosen are all within the Raspberry Pi family. The Zero, Zero W, and the Zero 2 W were selected. The top choices have been compared in terms of compatibility, ease of installation, and processing power. This comparison is presented in Table 3.

### 3.3. Realistic Use Cases for Appropriate Baseline Power Consumption

A large issue preceding the direct comparison from the power draw of an idling device to a device that is being attacked is the unrealistic usage scenario that would create. Devices within industries are likely to always have some sort of processing duties (in between standby/low power mode cycling). A few realistic use case scenarios could be based on initially connecting to the gateway and initialising the routing for the network, and then a series (or one) request can be cycled every so often to provide an update to the user. Multiple realistic usage scenarios have been outlined within research ranging from air quality monitoring systems [34] to critical health devices [35]. With multiple ranging baselines throughout the literature, average use is likely all dependent on the frequency of packet transfer of the devices. However, some sources claim 10 years at a data transmission rate of 128 bytes per 10 s [36] and others claim 306 h [37] on one charge with frequent data transfer (as it was a vital sign monitoring system). Therefore, it is important to select the best choice for an appropriate baseline power draw for the study, increasing the comparability of the study’s results to real-world examples where a default operation time has already been established. Basing the impact of any possible attacks on the decrease of operational time would likely result in an unbiased proposal for ranking the effectiveness of each propositioned attack.

## 4. Experimental Setup

This section provides a detailed description of the experimental setup that was undertaken to support our study. The section is broken down into individual experiments which encapsulate each experiment methodology and provide variables measured alongside reasoning. These experiments use the following hardware and software. Raspberry Pi Zero W; USB port voltage analyser based on the UM34C board, which supports accurate measurements and comes provided with an app that can be used to export graphs of data collected; Ubuntu 20.04 64-bit running on VirtualBox (version 6.1.34) on a system based on a Ryzen 7 3700X CPU with 32 GB of RAM; and Cooja, which emulates virtual Sky motes running pre-compiled Contiki-NG OS.

### 4.1. Experiment 1: DoS Attacks in a Simulated Environment

This subsection describes the attempted experimental setup of a DoS attack through a Tunslip interface with an IPv6 edge-node border-router network mote running Contiki-NG and tunslip6. After the installation and successful setup of the subsystems needed to launch Cooja, the following steps were attempted: creation of motes, verification of correct operation, SYN packet flooding, Ping of death, and finally UDP flood. The Sky motes which were selected for this experiment were running: rpl-border-router.c, which was tasked with being both the root node and connection into the network and udp-server.c, which would provide other nodes for interaction during experimentation. However, during the experiment, a multitude of scripting issues and incompatibilities were encountered, namely with Tun interface which only worked occasionally. The original scripting toolkit (Hping3) also did not work with the network as it was IPv6-based, and a scripting toolkit was utilised instead which worked with IPv6 [45].

The creation of motes was performed as follows:First, 9/10 motes are based on default RPL-compatible motes; these are Sky motes running Contiki-NG. These motes choose their own topology as they are utilising the RPL protocol. Hence, this can become a mesh or tree topology in the form of a destination-oriented acyclic-directed graph (DODAG). This corresponds to the simulation of a typical LoRa network;One mote is set up as a border router (IPv6), again using the default installation from the Contiki-NG package. Default values were used within the operating system for ease of use and deployment;Concurrent deployment of these motes within Cooja along with verification that they work correctly;Creation of a tun0 interface on the host OS ready to interface with the network edge mote;Initial ping of the border router from the host device through tun0-; when this is verified within Cooja, the experiments can start.

The attacks that were chosen were reply-based. This was carried out to create a clear picture of where a DoS is achieved, as the devices stop replying to packets. The attacks were made to be directly comparable on the physical scale to provide a clear vector of comparison between the two experiments. During the attempted experimentation, there were multiple variables to be measured. A packet analysis program was set up on the tunnel interface created and a .pcap file was to be recorded. Power usage was also measured as reported by the tools featured within Cooja. The attack focused on a disruption of normal service of the devices targeted. It was also draining the battery of the single point of communication within a network, which would lead to a single point of failure if the device’s battery is drained. The aforementioned experiment revealed that there was no communication possible through the tun0 interface. Therefore, it was decided to proceed with the DoS attacks on physical devices, as explained in Section 4.2.

### 4.2. Experiment 2: DoS Attack on Physical Devices

Given the outcome of the first experiment as explained in Section 4.1, our second experiment investigated the impact of DoS attacks on physical devices. This experiment follows the same attack methodology as the first experiment; however, it is a hardware-based experiment with a range of operating systems used. In other words, the second experiment was designed to be the physical counterpart of the first experiment. In particular, the hardware used was Raspberry Pi Zero and USB Power Analyser, as can be seen in Figure 1, while the software used was:Tiny core Linux (as close as possible to a bare-bones Linux installation);Yocto OS (an IoT-based distribution that may fail to compile for Pi Zero);Raspberry Pi OS (a Debian-based distribution to provide a possible contrast);Buildroot embedded Linux (a backup option for Yocto OS);Kali Linux (a backup option for Tiny Core Linux);Temp monitoring script;Benchmarking software;Balena-Etcher ISO to bootable media converter;Hping3 package.

**Figure 1 sensors-23-02605-f001:**
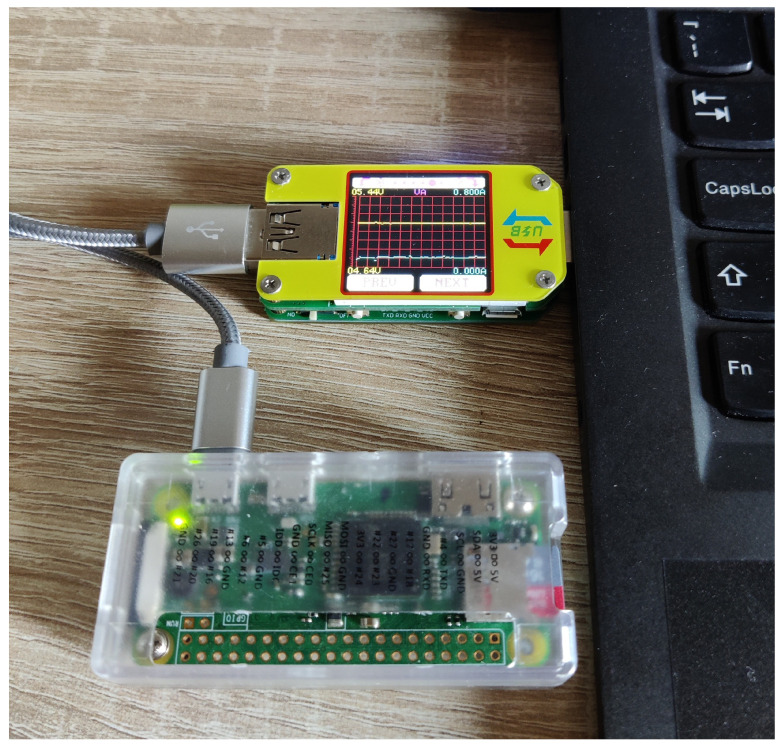
Physical setup of Experiment 2. The device in the clear case is the Raspberry Pi Zero W and the yellow USB device is the inline power draw analyser.

The same three-stage approach was used, similar to the first experiment; however, this time, the devices reside on the same physical network and without Internet connectivity. The attacks used the following default/well-known ports to keep the tests fair; they also did not use any ports above 1023 as those ports are usually not utilised. The UDP flood attack used port 80, SYN flood port 80, and ICMP flood targeted the ping functionality of the device which does not use a port. The estimated throughput of packets a second for each attack was: ICMP flood 5000/s, SYN flood 600/s, and UDP flood at 2500/s. In particular, the script was first run on Raspberry Pi OS, then on Tiny Core and finally on Yocto OS. However, due to technical issues, Raspberry Pi OS, Kali Linux, and Buildroot OS were used instead. During the experiment, the following variables were measured:A note of the device’s average core temperature is taken during normal operation and again during the attack to enable a vector of comparison between them;Power draw of the device during normal operation and under attack;Another PC intercepting network traffic and running Wireshark in mon0 mode.

A simplified version of this attack is presented in Figure 2. Wireshark was used for monitoring purposes. In total, three devices were connected to one network router, which was utilised as a switch and was used to enable the communication of the attacker and victim (Raspberry Pi Zero W), and the capture of packets sent between them to be able to calculate different attack loads that the Raspberry Pi can handle.

### 4.3. Experiment 3: RPL-Based Attacks

In this subsection, we present the initial steps of the creation of multiple benign nodes and a single malicious node, which were used in our implemented RPL-based attacks. This is common practice within simulation-based studies [46,47]. For simplicity, we disabled the communication with the host machine; this, however, can be changed to make different experiments where some interaction with the border router is needed. In our experiments, the network is isolated, and all interactions are with other motes only. The attack scenarios are explained below and are also presented in Figure 3, Figure 4 and Figure 5.

Rank attack: The malicious node reports a closer distance to the root node after the successful establishment of the RPL network. Within the network, most traffic from the tree nodes and surrounding nodes are routed through the malicious node. The aim of the attacker is to increase the power usage of devices and decrease the efficiency of the network.

DIS flooding attack: The network is flooded with as many RPL DODAG Information Solicitation (DIS) messages as possible. The motes that within reach send RPL DODAG Information Object (DIO) messages back. This behaviour is similar to a SYN–ACK response. The aim of the attacker is to drain the battery of the devices.

Version number attack: When forwarding DIO messages received from other motes, the version number is increased. This makes the other devices within the network rebuild bridges between the motes. This in turn triggers all repair messages within the RPL instance and causes large numbers of packets to be sent within the network.

The aforementioned attacks were implemented as follows:The network is set up with the majority of the motes being copies of a basic sensor with RPL capability;The motes are placed automatically, and randomly throughout the simulation with one root node and one malicious node;Before the attack is run, the option of creating a .pcap file is selected and the tracking of power consumption is enabled;The virtualisation of this network is commenced; it is run twice, once without the malicious node for a baseline, and once with the malicious node;When the attack is simulated successfully, it is stopped;The .pcap file is created and the measurements of the mote activity are collected: one before and one after the attack.

During the attacks, .pcap files of all communications are created, power draw is monitored, and DODAG graphs are analysed. We also increase the number of sensor nodes present within the network in order to simulate networks of different sizes. This impacts the DODAG routes and increases power consumption. In particular, we implemented three stages of the attack: a simulation with 10, 13, and 16 sensor nodes.

## 5. Results

This section presents and analyses the results of Experiments 2 and 3. Due to the nature of the results having a small sample size, any statistics derived from the data are susceptible to being influenced by extreme values within the data sets. From our collected measurements within Experiment 2, it can be seen that, during experimentation, there is a low point of data before the attack commences and, afterwards, including such data would heavily skew the accuracy of our results. This also applies to data within Experiment 3 as the small sample size will be susceptible to the same skewing. Steps were taken to consider this, most notably the trimming of the data set which is an important consideration from a statistical viewpoint to ensure that inferences from data are as accurate as possible. For comparison, we define the following attack levels depending on the percentage of the power draw increase: level-1 20%, level-2 40%, level-3 60%, level-4 80%, and level-5 100% or more.

In summary, the results of Experiment 2 show that DoS attacks are incredibly taxing on small, embedded-level systems. We observe a power draw increase for every attack and sometimes the attack even rendered the nodes non-responding. The most efficient system during the attack was Buildroot with only one attack reporting as a level-1 severity attack. This is a similar outcome to [48] where a similar concept of power draw analysis per timing interval was utilised, with the smoothing of the resulting data (normalisation). In particular, they utilised Arduino UNO devices with Dht22 sensors to investigate the potential decrease of power consumption if paired with machine learning algorithms. However, the resulting experimentation suggested that the more computational stress was placed on small-bandwidth nodes the higher their power consumption was, which was very similar to the outcome of this experiment.

Similarly, Experiment 3 shows that RPL-based attacks are also a significant danger to WSNs with most attacks increasing the overall power usage by more than 100%. Experiment 3 also focused on the up-scaling of networks to see if an increase in power draw was visible. This was proven to be true to a certain extent, with an average increase of power draw being larger in 13 sensor networks compared to 10 sensor networks. The data suggest that there is a certain ratio of sensors to malicious nodes where the most efficient attacks are possible; using data from the study, this ratio is likely 13 sensor nodes to 1 malicious node. A paper that analysed a similar conceptual model to the one utilised within Experiment 3 from an endpoint security perspective was [49], which revolved around different propositions to DDoS attack detection and recovery mechanisms. The paper analysed multiple propositions and found that the suggested FBDR method of DDoS attack mitigation outclassed other examples currently in use. The study in question also utilised the method of plotting time and power consumption to represent power draw over a set time period, whilst performing a set task. This closely relates to the methodology used within Experiment 3.

### 5.1. Experiment 2

Below, we present the results of Experiment 2, focusing on Raspberry Pi, Buildroot, and Kali Linux.

Raspberry Pi OS: The average baseline power draw of the device is calculated to be: 0.083 A. The max power usage is 0.144 A on average from the whole data set, and the trimmed data set is an average of 0.189 A. This is the theoretical maximum current draw of the device. Under the SYN flood attack, we have an average current draw (nominal data) 0.182 A, trimmed average 0.193 A, and an average temperature of 42.5 °C. Under the ICMP flood attack, we have an average current draw 0.102 A, trimmed average 0.104 A, and an average temperature of 46 °C. Under the UDP flood attack, we have an average current draw 0.083 A, trimmed average is not required, and an average temperature of 43.1 °C. The power (current) draw data are given in Figure 6 and the temperature data in Figure 7. Utilising the previously mentioned attack severity scores based on the percentage increase from baseline, each attack scored: UDP flood level-0, as it did not increase power draw; ICMP flood 25.3%, which is a level-1 attack; and SYN flood 132% increase, which is a level-5 attack.

Buildroot embedded Linux: The average baseline power draw of the device is calculated to be 0.116 A. The max power usage is 0.125 A on average from the whole data set, and the trimmed data set averages 0.150 A. This is the theoretical maximum current draw of the device. Under SYN flood, we have average current draw 0.109 A and trimmed average 0.111 A. Under ICMP flood, we have average current draw 0.137 A and trimmed average 0.142 A. Under UDP flood, we have average current draw 0.112 A and trimmed average 0.114 A. The current draw data for Buildroot are shown in Figure 8. Utilising the attack severity scores based on the percentage increase from baseline, each attack scored: UDP flood level-0, as it actually decreased the power draw of the device, which went down to −1.8%; ICMP flood 22.4%, which is a level-1 attack; and SYN flood: −4.4%, which similarly to the UDP flood actually decreased power draw, which again is level-0.

Kali Linux: The average baseline power draw of the device is calculated to be 0.089 A. The max power usage is a 0.123 A on average from the whole data set, and the trimmed data set is an average of 0.170 A. This is the theoretical maximum current draw of the device. Without tempmon.sh, we have the following: SYN flood average current draw 0.087 A and trimmed average 0.089 A; ICMP flood average current draw 0.139 A and trimmed average 0.168 A; and UDP flood average current draw 0.090 A and rimmed average 0.093 A. With tempmon.sh, we have the following: SYN flood average current draw 0.145 A, trimmed average 0.159 A, and an average temperature of 44 °C; ICMP flood average current draw 0.137 A, trimmed average 0.172 A, and an average temperature of 45 °C; UDP flood average current draw 0.123 A, trimmed average 0.159 A, and average temperature is not measured.

The power draw data for Kali Linux are shown in Figure 9 and the temperature data in Figure 10. Utilising the attack severity scores, we have: UDP flood level-0, as it only increased power usage by 4%; ICMP flood level-4 as it increased power usage by 88.7%; and SYN flood scored an attack level-0 as it did not increase current usage. Then, for attacks, whilst monitoring tempmon.sh, we have: SYN flood increased power draw by 78.6% which is a level-3 attack; ICMP flood: increased power usage by 93.2%, which is a level-4 attack; and UDP flood increased power usage by 78% over baseline, which is a level-3 attack.

The device that proved to be the most efficient was Buildroot OS as the simulations performed on the device resulted in the smallest current draw increase and, therefore, scored the best within the study. The reason why Buildroot OS has fared so well with the specific experiments (within this study) is likely because of its minuscule hardware requirements. An interpretation based on the data provided could point to an efficiency increase with a smaller number of drivers/ extra applications. Temperature results were also an important factor within testing; however, their importance may have been overemphasised as they do not tell much when it comes to a comparison standpoint. The current draw results are much more effective at providing sound conclusions when compared; however, they may have been more useful if all devices successfully exported them.

There are a number of attacks related to the current draw characteristics of WSNs. For example, a side-channel attack, where an adversary uses information obtained from the side-effects of a sensor, such as the power consumption or electromagnetic radiation [50], to obtain sensitive information, such as encryption keys [51]. In particular, these types of side-channel attacks can be performed by analyzing the power consumption patterns of nodes when performing cryptographic operations [52]. Such analysis of a sensor’s current draw allows an adversary to obtain information about the encryption keys being used in the network, allowing them to launch further attacks, such as eavesdropping on communication or altering the transmitted data [51]. To prevent side-channel attacks in WSNs, it is important to implement security measures that are designed to obscure the information obtained through side-channel analysis. This can be achieved through techniques such as power obfuscation, randomization, or by using specialized hardware that is designed to prevent side-channel attacks.

Furthermore, the current draw characteristics of WSNs can be used to disrupt the normal operation of the network by launching a jamming attack [53], where an adversary transmits high-power signals in the same frequency band used by the WSN, effectively blocking or interfering with the wireless communication between the nodes. Jamming attacks can have a significant impact on the normal operation of sensor networks. These impacts include disrupted or lost communication, and transmission of incorrect data based on faulty or manipulated information. In some cases, such attacks can take down the entire network.

Finally, power exhaustion attacks on WSNs can be the result of an adversary sending large amounts of data to one or more sensor nodes, effectively overwhelming the node and causing its power consumption to increase significantly [54]. Such attacks can cause the sensor to run out of battery power, potentially leading to a disruption in the normal operation of the network. This can have a severe impact on the performance and reliability of WSNs, especially if the network is composed of nodes with limited battery life. To prevent power exhaustion attacks, it is important to implement security measures that can detect and mitigate these types of attacks in real time [55]. This can be achieved through techniques such as rate limiting, where the amount of data that can be transmitted to a node is limited, or through the use of energy-efficient communication protocols that reduce the power consumption of the network [16]. Additionally, implementing encryption and authentication mechanisms can help to prevent unauthorized access to the network, which can limit the impact of a power exhaustion attack.

### 5.2. Experiment 3

Below, we present the results of Experiment 3. The following graphs are based on trimmed average power draw percentages for each attack. Each graph is set out with the number of nodes (10, 13, and 16) and a power draw average for a baseline network compared to the power draw of a network with a malicious node included.

Figure 11 shows the results of a blackhole attack. The 10-node attack resulted in a 3% power increase over baseline; this is a level-0 attack. The 13-node attack resulted in a 14.4% attack, which is also a level-0 attack. Finally, the 16-node attack resulted in power decrease of 13%, which is also a level-0 attack. Figure 12 shows the results of a DIS flooding attack. The 10-node attack resulted in a 641% increase in power draw from baseline, ranking a level-5 attack. The 13-node attack resulted in a 679% increase in power draw, again ranking it a level-5 attack. The 16-node network resulted in a 533% increase in power draw making it a level-5 attack. Figure 13 shows the results of a version number attack. The 10-node network achieved a 474% increase in power draw. The 13-node network increased power usage by 538%. The 16-node network increased power usage by 445%. In all these cases, we have a level-5 attack.

With two thirds of attacks being over the 100% power increase over baseline, the results are very one-sided with a majority of attacks posing a great danger to RPL-based networks. All attacks within the range have been identified as being level-5, and actually much higher than the threshold. The attacks within the study all reacted differently to the scaling up of networks. The black hole attack reacted best to a slight up-scaling of the network peaking at 13 nodes, providing a 14% power consumption increase over the baseline power consumption.

This result was likely due to the way the DODAG is formed around the malicious node, making the network lose efficiency. The DIS flooding attack also worked the most efficiently with the 13-sensor network model, providing a 679% increase in power draw. Loosing efficiency within a 16-node network was likely down to the ratio between attacking nodes to malicious nodes. Finally, the version number attack reported findings similar to other attacks, with an increased power usage by 538%. Using the data presented, there is inconclusive evidence to support the idea that all attacks increase efficiency when more devices are present in a network. However, there is data to support that the attacks have a specific node to malicious device ratio that allow them to function as efficiently as possible.

## 6. Discussion of Results

This section provides a high-level discussion on the experimental results and highlights the main lessons learned.

### 6.1. Experiment 2

The reason why Buildroot OS has fared so well with the specific experiments (within this study) is likely because of its minuscule hardware requirements. An interpretation based on the data provided could point to an efficiency increase with a smaller number of drivers/extra applications. Temperature results were also an important factor in testing; however, their importance may have been overemphasised as they do not tell much when it comes to a comparison standpoint. The current draw results are much more effective at providing sound conclusions when compared; however, they may have been more useful if all devices successfully exported them. Making conclusions from the data, it is likely that the most efficient device is based on Buildroot OS as can be seen in Figure 8. This is likely due to the low amount of overhead that the operating system needs to function; however, within the study, there is no consideration of systems working in an unintended way like freezing/not responding and due to having a low power consumption.

### 6.2. Experiment 3

With two thirds of attacks being over the 100% power increase over baseline, the results are very one-sided with a majority of attacks posing a great danger to RPL-based networks. All attacks within the range have been identified as being level 5, and technically a lot over the threshold to be level 5. This proves to be an issue as these results do not really mean much as not much comparison can be made between them. However, this is the reason why the percentage increases are also provided within the study, and these are going to be very important to make meaningful comparisons between the results. A way around this would have been to modify the scale and allow for more levels of attack, but this would likely detract from the other results that use a smaller scale. The results from this section were, however, not anticipated to be this large. The attacks within the study all reacted differently to the scaling up of networks. The black hole attack reacted best to a slight up-scaling of the network peaking at 13 nodes, providing a 14% power consumption increase over the baseline power consumption. This result was likely due to the way the DODAG graph of the network formed around the malicious node, making the network lose efficiency. The DIS flooding attack also worked the most efficiently with the 13-sensor network model, providing a 679% increase in power draw; losing efficiency within a 16-node network was likely down to the ratio between attacking nodes to malicious nodes. Finally, the version number attack reported similar findings to the two attacks before with an increased power usage by 538%. Using the data presented, there is inconclusive evidence to support the idea that all attacks increase efficiency when more devices are present in a network; however, there is data to support the attacks that have a specific node to malicious device ratio that allow them to function as efficiently as possible. Making conclusions from the data, it is likely that the peak network saturation is achieved at a ratio of 13 sensor nodes to one for most RPL-based attacks; this may directly be comparable to other attacks.

## 7. Conclusions

This paper explored the current draw potential of several types of attacks on embedded WSN devices that were represented by a Raspberry Pi Zero W. This device was selected because of a multitude of factors that make it an available platform to create and work with. The main reasons are the large community support, the price, and the availability of the device—which was an important factor due to the ongoing chip shortage in 2022. The device was subject to current draw analysis via an inline USB power draw analyser, which analysed a selection of operating systems running on the Raspberry Pi whilst being attacked by an adversary on a network. The device was subject to SYN flood attacks, ICMP flood attacks, and UDP flood attacks. The operating system that handled the attacks the best was Buildroot OS. Moreover, a virtual sensor network was established to probe the efficiency of RPL-based attacks on such devices. The networks were initially analysed by running without malicious nodes and then with them. During this simulation, the power draw of these networks was analysed and cross-compared to networks with malicious nodes. The malicious nodes utilised within the networks launched a selection of RPL-based attacks, notably the black hole, version number, and DIS flooding attacks. These attacks were scaled up to determine if an increase in power consumption will be noted, from 10 nodes to 13, and to 16. Results of each attack showed that there was an increase in power consumption when simulated within a 13-sensor large network. This suggests that there is likely a ratio of sensor nodes to malicious networks that works the most efficiently.

Regarding the methods chosen within this study, there were many positives that came from using averages within the Results section. Notably, many experiments could be seen drawing the maximum current draw of the device. If these results were considered only from their peak, many inferences from the results would have been lost. This method also maintained the accuracy of the results as it dealt with more data instead of, for example, power draw per packet as utilised within Grey et al.’s work [32]. This is again compared to the method employed by Jung et al. [33], which is evident within this study, constituting a higher accuracy of results.

The study successfully achieved to investigate and virtualise common attack concepts within RPL networks against WSN devices. It achieved this by implementing three RPL-based attacks within a Contiki OS-based virtualising provider—the Cooja simulator. We also created, compiled, and set up multiple operating systems suited for a physical embedded system. A standard measurement was used to enable the comparison of each system, which was achieved by utilising the current analyser during experimentation. Using these tools, the study was able to successfully demonstrate that operating systems do in fact differ in reaction to a selection of DoS attacks. Due to some limitations with experiment 1, there was no ability to compare the effects that common denial of service attacks had on sensor-based networks. However, this miscalculation was likely redacted with the potential of experiment 3 and the success of experiment 2, which still provided results comparable to common attacks within these networks. We also observed that devices are likely to be overwhelmed by the attack bandwidth and may stop computing packets, thus artificially limiting their current draw and biasing the results. This was likely only an issue with the UDP flood-type attack within the experiment and likely did not result in much of an overall change to results. This would be a recommendation for future research as it would guarantee a ‘level playing field’ for operating systems, discrediting potentially artificial outputs and disallowing them to skew the results.

## Figures and Tables

**Figure 2 sensors-23-02605-f002:**
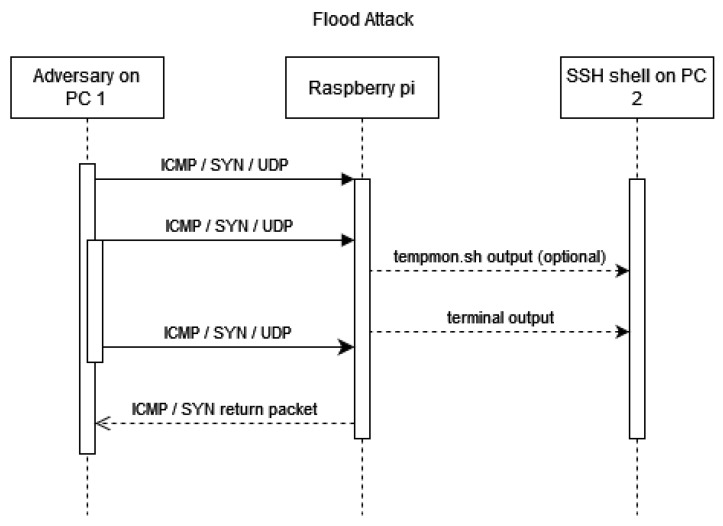
A simplified message flow diagram of all physical attacks. On the left is the adversary on PC 1, in the middle the Raspberry PI, and on the right is PC 2, which monitors the terminal output of the Raspberry Pi.

**Figure 3 sensors-23-02605-f003:**
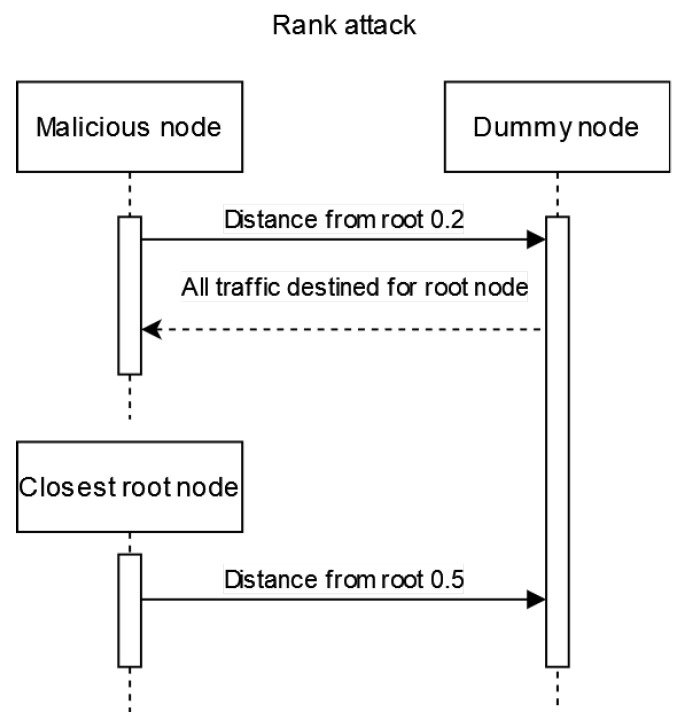
A simplified message flow diagram of a Rank attack in an RPL network.

**Figure 4 sensors-23-02605-f004:**
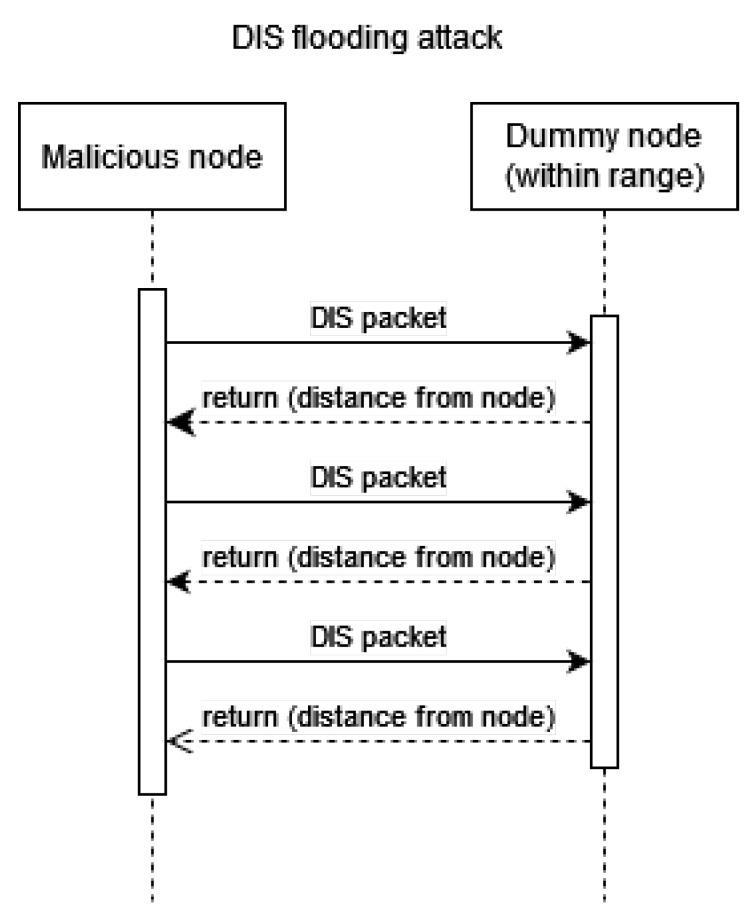
A simplified message flow diagram of a DIS flooding attack in an RPL network.

**Figure 5 sensors-23-02605-f005:**
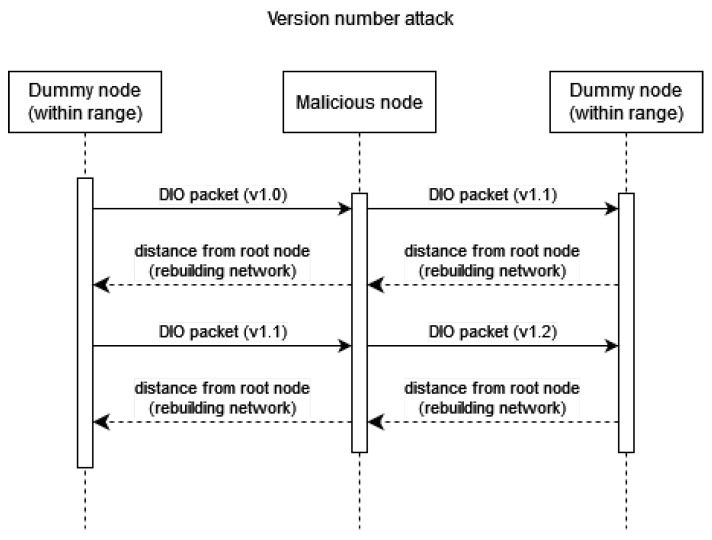
A simplified message flow diagram of a version number attack in an RPL network.

**Figure 6 sensors-23-02605-f006:**
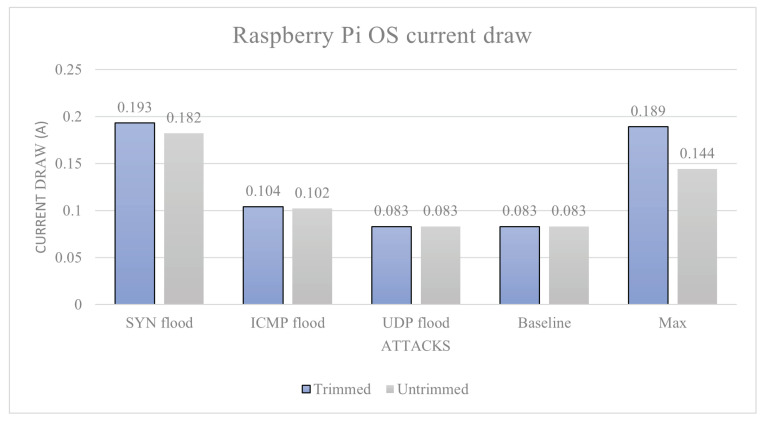
Raspberry Pi OS current draw.

**Figure 7 sensors-23-02605-f007:**
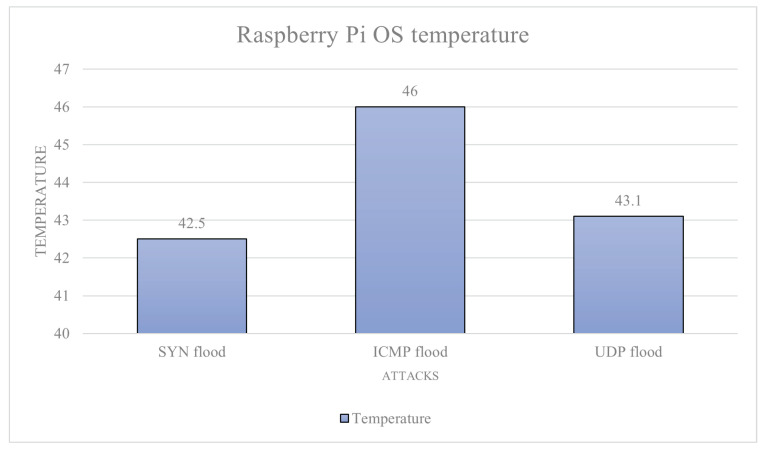
Raspberry Pi OS temperature.

**Figure 8 sensors-23-02605-f008:**
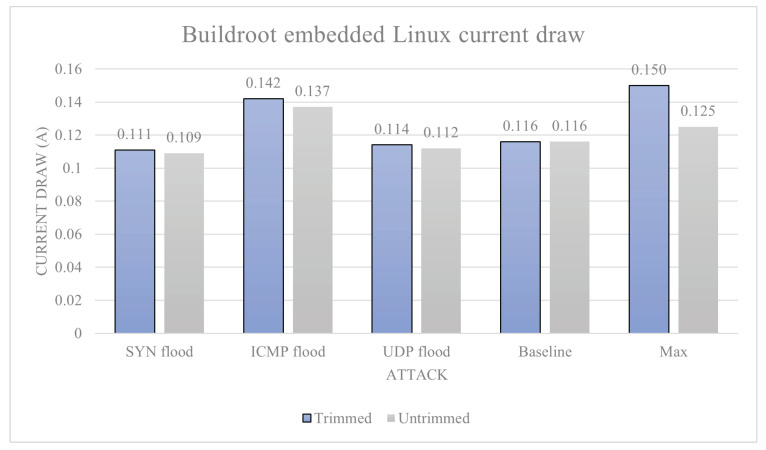
Buildroot OS current draw.

**Figure 9 sensors-23-02605-f009:**
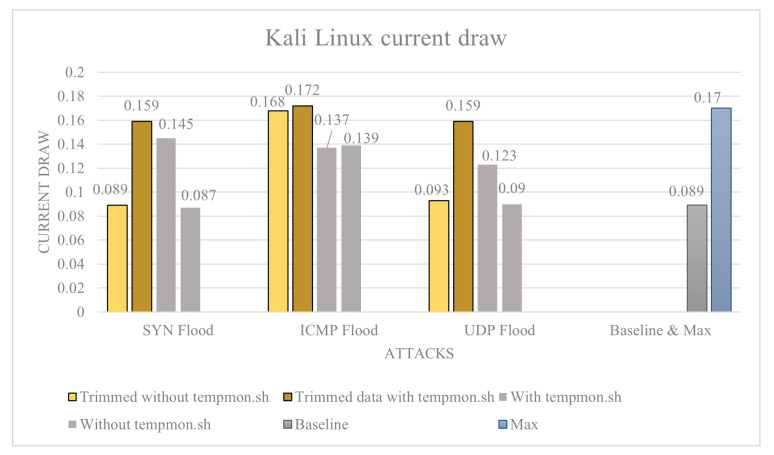
Kali Linux current draw.

**Figure 10 sensors-23-02605-f010:**
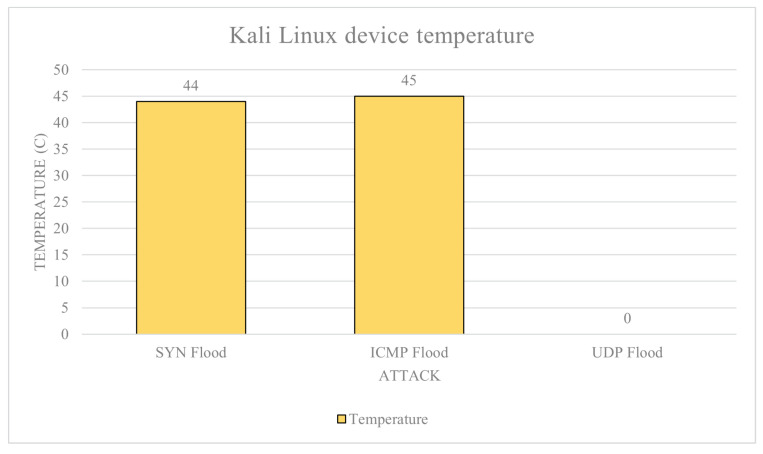
Kali Linux temperature.

**Figure 11 sensors-23-02605-f011:**
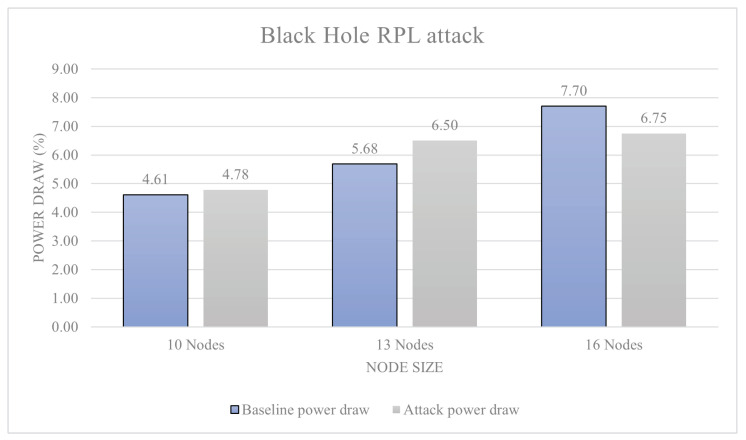
Black hole attack.

**Figure 12 sensors-23-02605-f012:**
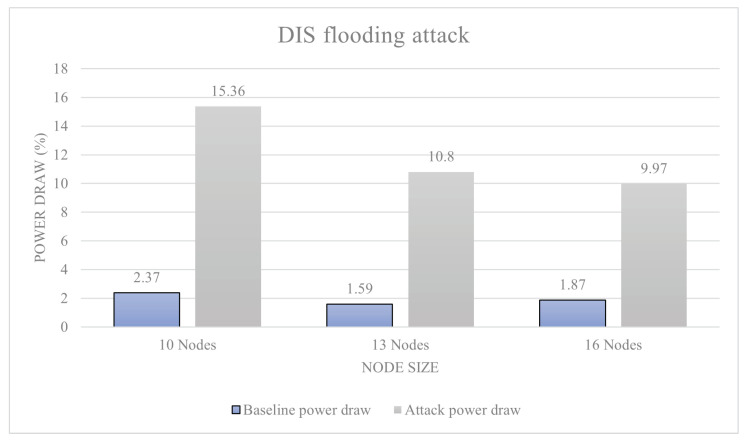
DIS flooding attack.

**Figure 13 sensors-23-02605-f013:**
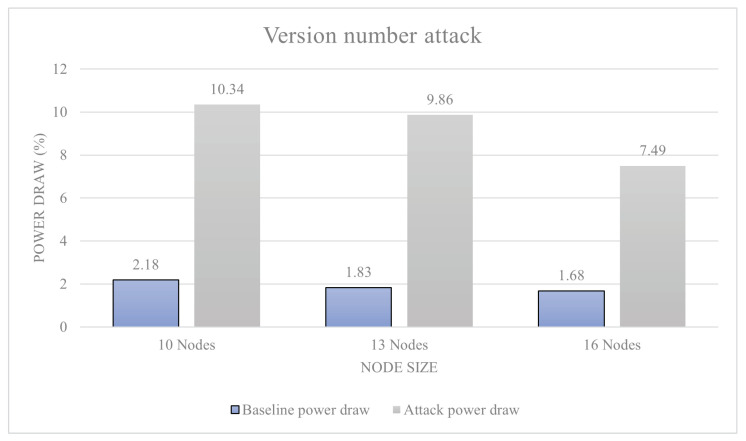
Version number attack.

**Table 1 sensors-23-02605-t001:** Operating systems available. Key: Y = Yes, N = No, S = Small, M = Medium, L = Large.

	Architecture Support (S, M, L)	Large Community Support (Y, N)	Low Power Protocol Support (S, M, L)	Kernel	Complexity (S, M, L)
Contiki-ng	S	Y	L	Contiki OS	M
RIOT OS	M	N	L	Unix	M
Free RTOS	L	N	L	RTOS	L
Ubuntu Core	M	N	L	Linux	S
Linux Arch	M	Y	S	Linux	S
Raspbian	S	Y	S	Linux	S
Tiny Core	L	Y	L	Linux	S
Buildroot	L	Y	L	Embedded	S
				Linux	
Android	S	N	S	Android	N/A
Things					
Yocto OS	L	N	L	Linux	L
Tiny OS	M	N	L	Component	L
				based	
Windows 10	L	N	L	Windows	S/M
IoT Core					
Azure	S	N	M	Linux	L
Sphere					
Kali	L	Y	S	Linux	S

**Table 2 sensors-23-02605-t002:** Hardware options available.

Hardware	Arch.	Large Community Support	Wireless Support	Price	Avail.	Extras	Support for OSes Selected
Intel	x86	Y	N	$50	N	Y	N
Edison							
Raspberry	ARM	Y	N	£5	Y	N	N
Pi Pico	(Armv6-M)						
Raspberry	ARM	Y	N	£10	Y	SD card	Y
Pi Zero	(Armv6)						
Raspberry	ARM	Y	Y	£12	Y	SD card	Y
Pi Zero W	(Armv6)						
Raspberry	ARM	Y	Y	£12	N	SD card	Y
Pi Zero 2 W	(Armv8-A)						
Zolertia	RISC	Y	Y	$170	N	SW	N
Z1							
Zolertia	ARM	Y	Y	50	Y	SW	Y
Firefly	(Armv7-M)						

**Table 3 sensors-23-02605-t003:** Hardware options compared for Raspberry Pi.

	Yocto OS	Tiny Core Linux	Raspberry Pi OS
Zero	Compatible. Medium ease of installation. However, lack of comparable results due to lack of wireless antenna. Some issues possible due to old processor.	Compatible. Medium ease of installation. However, lack of comparable results due to lack of wireless antenna. Some issues possible due to old processor.	Compatible. High ease of installation. However, lack of comparable results due to lack of wireless antenna. Some issues possible due to old processor.
Zero W	Compatible. Medium ease of installation. High likeliness of usable results. Old processor.	Compatible. Medium ease of installation. High likeliness of usable results. Old processor.	Compatible. High ease of installation. High likeliness of usable results. Old processor.
Zero 2 W	Compatible. Medium ease of installation. High likeliness of usable results. Modern processor.	Compatible. Medium ease of installation. High likeliness of usable results. Modern processor.	Compatible. High ease of installation. High likeliness of usable results. Modern processor.

## Data Availability

No new data were created or analyzed in this study. Data sharing is not applicable to this article.

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
