# Peer review of "An Analysis into Physical and Virtual Power Draw Characteristics of Embedded Wireless Sensor Network Devices under DoS and RPL-Based Attacks"

_sensors, 2023, doi:10.3390/s23052605_

Round 1

Reviewer 1 Report

1.     Missing a working flow chart of the algorithm.

2.     In addition to the measurement of indicators, it should be reflected in specific practical applications.

3.     The "Discussion" section is missing, it should be added.

4.      Some citations are very old.   Authors could add few updated and relative papers such as https://doi.org/10.1109/MNET.001.2100412;          Just to name some.

Author Response

Reviewer #1: 

  1. Missing a working flow chart of the algorithm. 

Authors’ response: Thank you for this comment. We have included a diagram of the physical experiment setup and algorithms of each experiment using message sequence diagrams. These are included in Figures 1-4  and explained in Section 4.

  1. In addition to the measurement of indicators, it should be reflected in specific practical applications.   

Authors’ response: In Section 5, we added more information about the specific applications of our considered indicators in different types of attacks on WSNs such as side-channel, jamming, and power exhaustion attacks. 

  1. The "Discussion" section is missing, it should be added. 

Authors’ response: We have added a new discussion section (Section 6) to this paper. This discussion provides more inferences on the results of the experiments, explanations for important results, and suggestions on how they could be improved.

  1. Some citations are very old.   Authors could add few updated and relative papers such as https://doi.org/10.1109/MNET.001.2100412;          Just to name some. 

Authors’ response: Thank you for the comment, we have included more papers that are relative to this area of WSN devices that are more up to date. 

Reviewer 2 Report

There is no information regarding the type of services running on the target machines during the TCP and UDP flood attacks. However, according to the type of service, there would be different responses when receiving a segment on their transport layer port.

Please Explain this sentence: Attacks are initiated from the host machine using python scripts hard-coded with the TCP sockets active in the border router.

The authors should specify the services provided by the nodes in experiment 1.

The authors must provide figures with the scenarios used in the experiments. This way, the figures in the scenarios will help those who read the work.

The work presents a scarce detailing of the types of attacks used. In addition, the authors must indicate the sending rate of the segments for the TCP flood and the UDP flood.

The experiments do not present novelty compared to others in the literature that use flooding with TCP/UDP/ICMP.

Author Response

Reviewer #2: 

1. There is no information regarding the type of services running on the target machines during the TCP and UDP flood attacks. However, according to the type of service, there would be different responses when receiving a segment on their transport layer port.

Authors’ response: Thank you for this remark. As minimal installations were used in experiment 2 many ports did not have running services on them. Conscious effort was made to use default ports (well known ones) to not colour results. In the received manuscript we provide the ports used in each attack and detail any services running on them in the manuscript to provide more information on this comment. Regarding experiment 1: we have clarified experiment 1’s shortcomings within Section 4.1 and made sure it was talked about in retrospect as it identified a number of technical issues that were taken into consideration in subsequent experiments. 

2. Please Explain this sentence: Attacks are initiated from the host machine using python scripts hard-coded with the TCP sockets active in the border router.

Authors’ response: We agree that this statement does not provide a clear picture given some limited information about the experimental setup in the original manuscript. In the revised version we have updated Subsection 4.1 to clarify this further and we have focused more on the two experiments that went ahead and produced sound results. 

3. The authors should specify the services provided by the nodes in experiment 1.

Authors’ response: Similarly to our response to the previous question, Subsection 4.1 was modified to remove this ambiguity. Basically, experiment 2 was a continuation of experiment 1 where we used physical devices to test DoS attacks. This has been clarified in the revised manuscript.

4. The authors must provide figures with the scenarios used in the experiments. This way, the figures in the scenarios will help those who read the work. 

Authors’ response: We have provided more figures (Figs. 1-5) to further explain the experiments within the paper. 

5. The work presents a scarce detailing of the types of attacks used. In addition, the authors must indicate the sending rate of the segments for the TCP flood and the UDP flood.

Authors’ response: All experiments now feature message flow diagrams explaining the data flow within them. We have also specified the sending rates of the attacks using the peak figures of each; this was done as the attacks used dynamic scaling and they were not done at a solid rate. The estimated throughput of packets a second for each attack was: ICMP flood 5000/s, SYN flood 600/s and UDP flood at 2500/s

6. The experiments do not present novelty compared to others in the literature that use flooding with TCP/UDP/ICMP.  

Authors’ response: Thank you for this comment. Although there are existing works that study the impact of following attacks on WSNs, our focus in this paper is specifically on the impact of these attacks on power consumption of sensor nodes. In addition, we also study other RPL-based attacks that do not use flooding, for example bakchole and version number attacks. In Section 4, we added more information about the experimental setup and additional diagrams to illustrate the implemented attacks. Finally, in Section 5 we added more information about the possible impact of attacks that exploit power draw characteristics of sensor networks.   

Reviewer 3 Report

The introduction and Related work are impressive by the big number of references to papers and it's obviously the research made for actual studies. Until the Experimental setup, all it looks like a big introduction. It will be good to present the methodology and some connectivity with an analytical model if it is possible.

The paper is much more a parallel between experimental data for the Rasberry Pi platform and it will be good to present also results in comparison with others platforms.

It can be seen the research is large but the structure of the presentation can be optimized for easy understanding and are presents some percentual values which will look better on graphs and in conclusions also are needed.

The conclusions can be improved  and it should present the advantages of the method and some comparative data reached after the study in percentages.

Author Response

Reviewer #3: 

1. The introduction and Related work are impressive by the big number of references to papers and it's obviously the research made for actual studies. Until the Experimental setup, all it looks like a big introduction. It will be good to present the methodology and some connectivity with an analytical model if it is possible. 

Authors’ response: Thank you for this suggestion. Our methodology is essentially covered in Sections 3 and 4. In Section 3, we explain our rationale behind the selection of specific operating systems and hardware platforms. In Section 4, a detailed methodology for our experimental setup and implementation is provided. We have clarified these points in the revised version. There are analytical models that describe the behaviour of power consumption attacks in WSNs. These models can be used to study the impact of such attacks on the performance and reliability of sensor networks. For example, queuing models, Markov chain models, and Petri net models. We agree that studying and developing such analytical models would be very useful, this however is beyond the scope of the current work which focuses on practical applications of the relevant WSN characteristics.

2. The paper is much more a parallel between experimental data for the Rasberry Pi platform and it will be good to present also results in comparison with others platforms.

Authors’ response: Thank you for your suggestion, we have provided a comparison to another study that revolved around the power draw analysis of WSN devices within the results section. It is also important to note that this is an area where future works are able to fill out the gaps in this study.

3. It can be seen the research is large but the structure of the presentation can be optimized for easy understanding and are presents some percentual values which will look better on graphs and in conclusions also are needed. 

Authors’ response: In the revised version, we updated Sections 4 and 5 with more graphs and explanations that clarify the considered attacks and the impact of WSN characteristics that were presented as part of our experiments. 

4. The conclusions can be improved  and it should present the advantages of the method and some comparative data reached after the study in percentages.

Authors’ response: Thank you for the comment, we have provided the advantages to utilising the method that was taken compared to the other works discussed in the paper. We have also added more information to the conclusion to make it more rounded overall.

Round 2

Reviewer 1 Report

No other problems

Reviewer 2 Report

The authors have satisfactorily addressed most of my concerns.